# Phylogenetic Characterisation of the Full Genome of a Bagaza Virus Isolate from Bird Fatalities in South Africa

**DOI:** 10.3390/v14071476

**Published:** 2022-07-05

**Authors:** Adriano Mendes, Olivia Lentsoane, Mushal Allam, Zamantungwaka Khumalo, Arshad Ismail, Jacobus A. W. Coetzer, Marietjie Venter

**Affiliations:** 1Zoonotic Arbo- and Respiratory Virus Research Program, Centre for Viral Zoonoses, University of Pretoria, Pretoria 0007, South Africa; u18387293@tuks.co.za (A.M.); olivialentsoane@gmail.com (O.L.); 2Sequencing Core Facility, National Institute of Communicable Disease, Johannesburg 2192, South Africa; mushala@nicd.ac.za (M.A.); zamantungwak@nicd.ac.za (Z.K.); arshadi@nicd.ac.za (A.I.); 3Department of Veterinary Tropical Diseases, Faculty of Veterinary Science, University of Pretoria, Pretoria 0001, South Africa; koos.coetzer@up.ac.za

**Keywords:** flavivirus, Bagaza virus, full genome sequence, Israel turkey meningoencephalomyelitis virus, phylogenetics

## Abstract

Bagaza virus (BAGV), a member of the Ntaya serogroup in the *Flavivirus* genus of the *Flaviviridae*, was isolated from the brain tissue of a Himalayan monal pheasant that died following neurological signs in Pretoria, South Africa in 2016. Next-generation sequencing was carried out on this isolate resulting in a genome sequence of 10980nt. The full genome sequence of this isolate, designated ZRU96-16, shared 98% nucleotide identity with a BAGV isolate found in *Culex univitattus* mosquitoes from Namibia and 97% nucleotide identity with a Spanish BAGV sequence isolated from an infected partridge. In total, seven amino acid variations were unique to ZRU96-16 after alignment with other BAGV and Israel turkey meningoencephalomyelitis (ITV) genomes. The 3′UTR sequence of ZRU96-16 was resolved with sufficient detail to be able to annotate the variable and conserved sequence elements within this region. Multiple sequence alignment of the 3′UTR suggested that it could be useful in lineage designation as more similar viruses carried similar mutations across this region, while also retaining certain unique sites. Maximum likelihood phylogenetic analysis revealed two clusters containing both BAGV and ITVs from Europe, the Middle East and Africa. Broadly, temporal clustering separated isolates into two groups, with one cluster representing viruses from the 1960–2000’s and the other from 2010 onwards. This suggests that there is consistent exchange of BAGV and ITV between Europe and Africa. This investigation provides more information on the phylogenetics of an under-represented member of the *Flaviviridae* and provides an avenue for more extensive research on its pathogenesis and geographic expansion.

## 1. Introduction

The *Flaviviridae* are a family of positive-sense single-stranded RNA viruses known to cause outbreaks of human and animal disease. The family consists of four genera: Hepaci-, Pegi-, Pesti- and Flavivirus [1]. The genus Flavivirus currently contains 53 recognised species, primarily transmitted via arthropod vectors [2,3]. Bagaza virus (BAGV) is a re-emerging Flavivirus, transmitted to birds, in particular turkeys and pheasants in the *Phasianidae* family, by Culex mosquitoes. It was first discovered in the Bagaza region of the Central African Republic in 1966 [4]. From the 1960s onwards, BAGV has been identified consistently in mosquito pools in West Africa [5,6,7]. The virus was also isolated from a single pool of *Culex tritaeniorhynchus* mosquitoes in India together with serological evidence of human exposure in hospitalized encephalitic patients [8]. In 2010, the virus was responsible for an outbreak of neurological disease in red-legged partridges (*Alectrois Rufa*) and common pheasants (*Phasianas colchcius*) in Spain [9]. During this outbreak, all 13 of the dead birds were found to be PCR positive for the virus [9]. Like West Nile virus (WNV) to the Americas [10] and Japanese encephalitis virus (JEV) to Australia [11], this signalled an expansion of the host range of BAGV, to Europe. This was also the first time BAGV was confidently associated with disease in vertebrates. Another epornitic Flavivirus, Israel turkey meningoencephalomyelitis virus (ITV) causes a similar disease in turkeys. ITV has only been identified in Israel and South Africa, and causes fatal neurological disease in domestic turkeys (*Melagris gallipavo*) [12,13]. ITV is severely neuropathogenic, with mortality rates of between 15 and 30% and morbidity of approximately 80% [14]. Following the identification of BAGV in Spain, ITV and BAGV have been proposed as the same species, owing to cross-neutralization and they’re having over 90% sequence similarity; however, differences in geography, vector, host, disease association and ecology have led the International Committee on Taxonomy of Viruses (ICTV) to consider them as separate [14,15]. Our group reported an outbreak of BAGV in South Africa in Himalayan monal pheasants (*Lophophoius impejanus*) in 2016–2017 [16]. BAGV was identified over three consecutive years in these exotic birds (a total of eight PCR-positive cases) in the same area, resulting in fatal neurological infections. This was the first time since 1980 that a virus resembling ITV or BAGV had been detected in South Africa. The potential of BAGV as a zoonotic pathogen as well as its impact on agriculture raises the significance of this introduction. 

To date, 20 complete or near-complete BAGV genomes have been uploaded onto GenBank. However, geographically and in terms of the origins of the isolates, there is limited diversity in these genomes. Twelve genomes are from a similar region in West Africa, mostly from Senegal [17]. Five sequences are from Spain, which appear to originate from a single isolate [9,18]. Single genome sequences from India, Zambia and Namibia are also available, but these are the only representatives that expand the geographic range outside of West Africa and Spain [19]. Of the 20 genomes, all but the Spanish isolate were derived from Culex mosquitoes and were sequenced using overlapping PCR segments and Sanger sequencing. There are even fewer ITV whole genome sequences available. There are currently only 6 full genome sequences on GenBank all of which were isolated from infected turkeys in Israel. Here we describe a complete BAGV genome derived using deep sequencing. To our knowledge, this is the first full genome sequence of BAGV from South Africa and the second BAGV genome from a virus isolated from the brain of an infected bird. This sequence prompted us to re-examine the phylogenetic and temporal relationship of BAGV and ITV isolates.

## 2. Materials and Methods

### 2.1. Virus Isolation and RNA Extraction

The BAGV isolate ZRU96-16-2_Monal_South_Africa was identified from the brain tissue of a Himalayan Monal Pheasant (*Lophophoius impejanus*) that died following neurological signs, submitted to the Centre for Viral Zoonoses at the University of Pretoria, South Africa in April 2016. The specimen tested positive for a flavivirus using a pan-Flavi PCR and was subsequently identified by Sanger sequencing as BAGV [20,21]. The original identification of this specimen formed part of an initial study describing an outbreak of BAGV in pheasants on a private residence in Pretoria [16]. All virus manipulation and nucleic acid extractions were performed under Biosafety level 3 conditions. The isolate was obtained from two rounds of serial passage of the homogenized brain tissue in BHK-21 cells (ATCC^®^, Manassas Historic District, VA, USA), maintained in Eagles minimal essential media (EMEM) (Lonza; Basel, Switzerland) with 5–10% fetal bovine serum (FBS) (Gibco; Leicestershire, UK), and Mycozap (Lonza; Basel, Switzerland) at 37 °C with 5% CO_2_, until 80% cytopathic effect (CPE) was observed. The supernatant from the second passage stock was used to inoculate 3× T75 flasks of the same cells for five days. The cells and supernatant were then subjected to a series of three freeze-thaw cycles at −80 °C to release any cell associated virus and then centrifuged to remove cellular debris. The resulting supernatant was passed through Minisat single-use 0.45 filter columns (Sartorius Stedim Biotech; Gottingen, Germany) to remove any possible bacterial contamination followed by further filtration using Millipore Amicon Ultra-15 Centrifugal Filter Units 10 K (Merck; New York, NY, USA) at 5000× *g* for 40 min until the supernatant was concentrated to a volume of 500–600 µL. BAGV RNA was then extracted using the Direct-zol RNA Miniprep kit (Zymo Research, Irvine, CA, USA) according to the manufacturer’s instructions. The RNA was further cleaned up using an RNA Clean and Concentrator^TM^-5 kit (Zymo Research, Irvine, CA, USA) to remove all fragments smaller than 200 bp to a final volume of 15 µL prior to cDNA synthesis.

### 2.2. Sequence-Independent Single-Primer Amplification (SISPA) and Rapid Amplification of cDNA Ends (RACE) for Deep Sequencing

For high-throughput sequencing (HTS), cDNA was generated using a SISPA-RACE protocol as described in [22]. Briefly, Poly-U RNA was generated by adding 7 µL of the purified RNA template to a mixture of 2 µL 10× NEBuffer (New England Biolabs, Ipswich, MA, USA), 1 µL of 100 mM uridine 5′-triphosphate (UTP) (Thermo Fisher Scientific; Waltham, MA, USA), 1 µL RNAse inhibitor (Thermo Fisher Scientific, Waltham, MA, USA) ), 1 µL Poly-U polymerase (New England Biolabs, Ipswich, MA, USA) and 8 µL nuclease free water to a final volume of 20 µL and incubating for 10 min at 37 °C followed by quick chilling on ice. A volume of 7.5 µL of the generated poly-U RNA was then added to a mixture of 2 µL of 10 µM FR-26-RV-N primer (5′-GCCGGAGCTCTGCAGATATCNNNNNN-3′), 2 µL of 1 µM FR40RV_anchored A primer (5′-GCCGGAGCTCTGCAGATATCAAAAAAAAAAAAA-3′) and 0.5 µL of 50% DMSO (Thermo Fisher Scientific, Waltham, MA, USA) to a total volume of 12 µL. This reaction was incubated for 5 min at 65 °C, 2 min at 4 °C, and subsequently placed on ice.

First strand cDNA was synthesised by adding 4 µL 5X RT Buffer Maxima H (Thermo Fisher Scientific, Waltham, MA, USA), 1 µL of 100 µM Venter SISPA primer (5′-GCCGGAGCTCTGCAGATATCGGCCATTATGGCCGGG-3′), 1 µL Maxima H-RT (Thermo Fisher Scientific, Waltham, MA, USA), 1 µL of 10 mM ddNTPs (Invitrogen) and 1 µL RNAse inhibitor (Thermo Fisher Scientific, Waltham, MA, USA) to the 12 µL denatured Poly-U RNA reaction and then incubating for 10 min at 20 °C, 30 min at 50 °C, 15 min at 65 °C, 5 min at 85 °C. This was followed by second strand synthesis by adding 1 µL of RNAse H (Invitrogen; Carlsbad California, United States) and 1 µL of Klenow 3-5′ exo DNA polymerase (New England Biolabs, Ipswich, MA, USA) to the first strand synthesis mix and incubating for 60 min at 37 °C and 15 min at 75 °C. The product was then subjected to cDNA clean up using the PCR Mini Elute kit as per manufacturer’s instructions (Qiagen; Hilden, Germany).

The cDNA was amplified using a random fragment amplification protocol by a single primer (10 µM FR-20-RV: 5′-GCCGGAGCTCTGCAGATATC-3′) and Taq PCR red mix (Ampliqon) according to the manufacturer’s instructions. The cycling conditions were 98 °C for 30 s for 1 cycle, 98 °C for 10 s, 54 °C for 20 s, 72 °C for 45 s for 40 cycles and 72° C for 5 min. A second amplification was carried out using the same reaction conditions for a further 5 cycles. The product was purified for HTS using the Qiagen MiniElute kit. Paired-end libraries (2 × 300 bp) were prepared with the Nextera DNA Flex library preparation kit (Illumina, San Diego, CA, USA) and sequencing was performed on an Illumina MiSeq instrument by the National Institute of Communicable Diseases Sequencing Core Facility, South Africa.

### 2.3. Bioinformatic and Phylogenetic Analysis

Assembly of the ZRU96-16-2_Monal_South_Africa full genome and sequence analysis was carried out using the CLC Genomics Workbench 12 software package (Qiagen; Hilden, Germany). Firstly, the quality of the reads was assessed using the QC for sequencing reads tool. Since the read length was significantly enriched for reads between 295 and 304 bp, there were very few ambiguous bases, and 95% of the data contained PHRED scores of between 35 and 40, it was unnecessary to apply any quality filtering to the dataset. Reads were mapped to the Bagaza reference genome: DakAr B209 (NC_012534) as well as Bagaza isolate BAGV/Spain/RLP Hcc3/2010 complete genome (KR108245) employing a standard linear gap cost. A local realignment was performed on the mapped reads as well as a low frequency variant calling analysis. Additionally, a de novo assembly method was employed by submitting the forward and reverse reads to the Genome Detective online mapping tool [23]. Consensus sequences from reference mapping and de novo assembly were compared and the only difference was found at the beginning and end of the sequences. Since the de novo method yielded greater sequence information at the ends, this was used as the final consensus. The consensus sequence, designated ZRU96-16-2_Monal_South_Africa, hereafter referred to as ZRU96-16, was annotated using the amino acid cleavage sites described in [24] using CLC Main Workbench 20. All multiple sequence alignments and subsequent consensus sequences were also generated and annotated in CLC using only full genomes of BAGV or ITV published on GenBank. Alignments for phylogenetic analysis were generated in MAFFT [25]. Maximum likelihood analysis using a general time reversible model with discrete gamma distribution and invariable rate variation (GTR+G+I) was carried out in MEGA 7 [26].

## 3. Results

### 3.1. South African Bagaza Virus Isolate ZRU96-16-2_Monal_South_Africa Complete Genome Annotation

From a total of 221,146 reads, 138,627 (62.69%) mapped to the Bagaza reference strain DakAr B209 (Genbank accession: NC_012534), resulting in an average coverage of approximately 3700×. De novo assembly resulted in a single contig that covered 98% of the genome. The only major difference we observed between generating a consensus genome from reference mapping compared to de novo assembly was in the resolution of the 3′ end of the genome. De novo assembly resulted in an additional 134 nt at the end of the genome. To establish whether this was due to the process of reference mapping itself or the genome to which the reads were referenced, i.e., the DaKar B209 sequence, we carried out another reference mapping experiment, this time to the BAGV Spanish isolate (KR108246). We found that although the coding sequences of the genome remained unchanged, reference mapping to a different genome, resolved a greater area of the 3′UTR, as was the case after de novo assembly. The de novo contig was considered the final consensus genome for our isolate.

The final assembled genome was designated ZRU96-16-2_Monal_South_Africa, but will be referred to throughout this article as ZRU96-16. The consensus genome is 10,960 nucleotides in length. Using the cleavage sites published in Kuno and Chang 2007 [24], the canonical C-PrM-E-NS1-NS2-NS3-NS4A-2K-NS4B-NS5 gene annotation was assigned (Appendix A). The size of each gene matched that of the reference except for NS4B which was predicted to be 1 amino acid (aa) larger. The first nucleotide, which is generally an adenine, could not be resolved on ZRU96-16, and thus it begins with a guanine. The 3′UTR of ZRU96-16 is 17nt longer than the Dakar reference strain but 33nt shorter than the Spanish isolate KR108246.

ZRU96-16 (GenBank accession: MW463911) was compared to all other BAGV and ITV full genome sequences available on GenBank using multiple sequence alignments (MSA). A selection of the alignment illustrating only the variant sites across the polyprotein is depicted in Figure 1. In total, seven amino acid variations were identified as being completely unique to ZRU96-16. Two (28.5%) of these differences were found in the structural genes and Five (71.5%) within the non-structural genes. Within the non-structural genes, NS4B and NS5 were the most variant genes, with two unique sites each (Figure 1). The amino acid mutations R2768K and T2973I within NS5 were unique to ZRU96-16, while the G3048S was only present on the South African and Namibian isolates. This was of interest, as NS5 is generally highly conserved. The majority of the amino acid variation was derived from the West African sequences NC102534 and MF380425 or the Indian isolate EU684972 (Figure 1). This was characterized by stretches of unique amino acid sites in Capsid, PrM, NS3 and NS4B.

Pairwise comparison of the 20 BAGV full genomes at the nucleotide level showed ZRU96-16 to have the highest sequence similarity to the most recently discovered isolate from Namibia (MW672101) at 98.21% similarity (P distance 0.01). The isolates from Spain and Zambia were the next most similar sequences (percent identity between 95 and 97%, P distance 0.024 and 0.023) (Appendix A). The isolate from India was the least similar to ZRU96-16 and any other isolates at between 88 and 89% identity. All the BAGV sequences had similarities ranging between 97 and 100% at the amino acid level. All six ITV genomes were highly similar to one another at both the nucleotide (92–99% identity; P distance 0 to 0.06) and amino acid level (98 and 99% similarity). ZRU96-16 showed the highest similarity to two of the five ITV full genome isolates at 94.99 (KC734553) and 94.92% (KC734551) similarity (P distance of 0.036).

### 3.2. Sequence Analysis of the 3′UTR Sequence of ZRU96-16

Upon analysis of the complete MSA, we noted several nucleotide variations over a short stretch of the genome. This fell within the 3′UTR, and thus we decided to characterize this portion of the genome on the South African isolate versus other BAGV and ITV sequences. It is important to note that when we carried out reference mapping against the BAGV reference strain (NC_012534), the 3′ end terminated at position 10,811, resulting in a large truncation of the genome. However, de novo assembly revealed that this was an artifact of the assembly process, and that ZRU96-16 was in fact 10,960 nt in length. Many arboviruses contain within their genome sequences conserved sequences, elements/motifs (CS) that are important regulators of viral RNA recognition during RNA replication, packaging or assembly. Flaviviruses in particular have well characterized 3′ UTR regions that are evolutionarily constrained in order to maintain specific secondary structures [27]. By annotating the alignment with the conserved sequence motifs and their repeats, identified specifically for BAGV (CS1-3), we observed that as has been previously reported, BAGV and ITV 3′ UTRs could be divided into a variable region, extending from the stop codon for approximately 178 nt, as well as a conserved region making up the rest of the genome. The variable portion is characterized by multiple variant sites, some of which are conserved among different isolates and others which are unique to a specific isolate (Figure 2). The conserved region of the UTR is characterized by conserved sequence motifs. The canonical order in which the conserved motifs has been described in BAGV is repeatCS3-CS3-repeatCS2-CS2-CS1 [28]. The order of these motifs was maintained in the BAGV and ITV full genomes that we analysed (Figure 2). Exceptions were in the case of the Zambian BAGV (LC318701) and all the ITV sequences, where sections after the CS2 were not resolved. The BAGV reference sequence was also an exception, as it contained a gap over which the CS2 was located on other sequences. Broadly, the 3′UTR MSA revealed two groups of BAGV and ITV sequences: the West African sequences, which shared greater similarity (94.5–99.8% percent identity), and the South African, Namibian, Spanish and Israeli sequences which were more similar to each other (97.5–99.7% sequence similarity). This is best exhibited in the CS1 sequence, in which the South African, Namibian and Spanish sequences all share a CS1 identical to that of the original reference strain. The West African sequences have the same 11 variant sites within this sequence. A similar trend is seen in the RCS2. The CS2 and CS3 were completely conserved.

### 3.3. Phylogenetic Analysis of ZRU96-16 within the Flavivirus Ntaya Antigenic Complex

A maximum likelihood phylogenetic tree was constructed for the mosquito-borne Ntaya virus group. The tree was generated using all of the complete genomes of BAGV and ITV available on GenBank as well as those of Ilheus virus (ILHV), Rocio virus (ROCV), Ntaya virus (NTAV) and Tembusu virus (TMUV). NTAV, TMUV, ILHV and ROCV formed distinctive separate lineages from those of BAGV and ITV (Figure 3). This provided confidence in the tree, as these viruses have previously been reported to cluster separately within the Ntaya complex [2]. After diverging from NTAV, both BAGV and ITV followed the same ancestral lineage. The ancestral virus then appeared to diverge into two clades, each containing highly similar BAGV and ITVs. The first clade consists of sequences derived between the 1960s and 2000s, while the second consists of those from 2010 to 2018, when our isolate was sequenced. The first clade contains closely related West African isolates which cluster independently from the Indian isolate (EU684972), which in turn form a group separate from ITV species. ZRU96-16 could be found in the second clade and shared a most recent common ancestor with the Namibian BAGV isolate (MW672101), collected from *Culex univitattus* mosquitoes in 2018. These isolates shared a common ancestor with another southern African sequence from Zambia (LC318701). These isolates were in turn closely related with the Spanish BAGV isolates, collected from Pheasants in 2010, which themselves were nearly indistinguishable. ITV species isolated in 2010 were also part of this second grouping. Overall, phylogenetic analysis revealed that the ZRU96-16 isolate is most closely related to other Southern African isolates from Namibia and Zambia, but also shares ancestry with more recently isolated BAGV and ITV’s from Europe and the middle east (Spain and Israel).

## 4. Discussion

Between 2016 and 2018, several Himalayan monal pheasants (*Lophophoius impejanus*) kept on private residences in Johannesburg succumbed to neurological disease from what was confirmed to be Bagaza virus infections. Phylogenetic analysis from partial NS5 gene regions showed that the outbreak was caused by a distinctive set of BAGV strains most closely related to Spanish BAGV sequences [16]. Here, we describe the full genome sequence of a South African BAGV virus isolate, ZRU96-16, which was isolated using tissue culture from the brain homogenate of one of the dead pheasants in 2016 and was sequenced using deep sequencing technology. The full genome sequence of ZRU96-16 showed it to be a unique Bagaza virus isolate in the flavivirus genus. This was due to the fact that the sequence contained, among several single-nucleotide changes, seven that resulted in unique amino acid changes on the polyprotein (Figure 1). The amino acid variations within the NS5 protein were of interest, as generally this region is well conserved between flaviviruses. However, despite this we observed several locations in which variations were accommodated across the different BAGV and ITV sequences. This may be explained by the amino acid forming part of a portion of the RNA-dependent RNA polymerase (RdRp) which is not structurally constrained and/or the fact that there are still very few genome sequences from these viruses and thus the full breadth of the diversity in this region has not been resolved. This is exemplified by the fact that the G3048S variation would have been unique if not for the relatively new Namibian BAGV sequence.

There were several additional variant nucleotides within the 3′ untranslated regions (UTR) (Figure 2). This was interesting, as the 5′UTR was very well conserved (data not shown). After annotation of the 3′UTR sequence, a clear pattern was seen in which geographically and temporally similar isolates had similar UTRs. The 3′UTRs from the West African isolates from 1988 to 2000, shared differences within the variable region as well as a conserved set of variations within the CS1. The South African, Namibian and Spanish sequences, all isolated from 2010 onwards also shared similarities within the variable region and a more canonical CS1. This may be evidence that the 3′UTR of these viruses is able to accumulate mutations which allow it to take advantage of a particular geographical niche.

The UTRs of several flaviviruses have been studied in detail. These regions of the genome contain conserved RNA secondary structural elements that are responsible for protein-RNA recognition during the replication cycle as well as protection from degradation. The conserved sequence elements (CS) form part of dumbbell and stem loop structures, which have been identified as essential for replication of mosquito-borne flaviviruses such as West Nile virus (WNV), Dengue virus (DENV) and Zika virus (ZIKA) [29]. DENV, for example, has been shown to require several RNA secondary structural elements at both the 5′ and 3′ end that are necessary for initiation and enhancement of replication [30]. Stretches of the 3′UTR of DENV have been identified that are under specific selection pressure according to the host that the virus is replicating in [28,31]. Specifically, mutations altering a small hairpin structure within the 3′UTR of DENV abrogated replication within insect cells with minimal effects in BHKs, illustrating the different requirements between hosts [32]. The greatest variation within BAGV and ITV 3′UTRs was observed in the region proximal to the stop codon of the polyprotein (Figure 2). This region is referred to as domain 1 and has been demonstrated to contain the pseudoknot sequences responsible for subgenomic RNA generation by incomplete exoribonuclease degradation [29]. Interestingly, two versions of the BAGV CS1 were found after alignment. CS1 has been shown to play a critical role in the cyclization of WNV RNA, which is required for RNA replication. Unlike WNV, the BAGV genomes did not contain insertions or deletions in the 3′UTR, which could assist in lineage designation [19]. The nucleotide sequence of the variable region may be of use in this regard for BAGV and ITV.

Phylogenetic analysis of the Ntaya antigenic complex of Flaviviruses with the new member from South Africa showed that ZRU96-16 clustered with the most recently isolated BAGV and ITVs (Figure 3). It shared a most recent common ancestor with a Namibian isolate from 2018 and then a Zambian isolate from 2013, as well as the Spanish isolates from 2010. It was more distantly related to the West African isolates. The ITV isolates, although still considered as a different species, from 2010 were more closely related to the African and Spanish BAGV isolates than the West African BAGV isolates. Both clades within our phylogenetic tree contained European, Middle Eastern and African species, thus suggesting that there was exchange of viruses between these regions. The fact that the phylogenetic tree seems to have been split by the time at which the isolates were found may suggest that this exchange is ongoing. Migratory birds would be the most likely transmission source for virus exchange, as the natural host birds—pheasants, partridges and turkeys—are ground dwelling. Many migratory birds such as swallows, storks, cranes and buzzards utilize routes that either take them from Southern Africa or West Africa through Southern Europe (Italy or Spain) or the Middle East in order to get to China and Russia [33]. These species would thus provide plausible intermediate hosts for BAGV and ITV. Phylogenetic analysis also illustrated that BAGV and ITV share lines of ancestry, and given the fact that Spanish isolates and now a South African isolate have a similar 3′UTR structure to all known ITV genomes, may provide sufficient evidence that it is time to reconsider the classification of these two viruses as separate species.

## 5. Conclusions

BAGV and ITV are lesser-known flaviviruses when compared to viruses like DENV, WNV and Zika, but are associated with fatal outbreaks in pheasants and turkeys in Spain, Israel and now also South Africa. Reports in India suggested association with encephalitis in humans [8]. Further investigation is needed to determine the zoonotic potential in other areas where they have been detected. The short viremia associated with many flaviviruses may necessitate such studies to be serological in humans, with a focus on febrile and neurological cases. This study provides insight into the phylogenetic and molecular epidemiological relationship of Bagaza strains identified across continents and over several decades suggesting temporal as well as geographic clustering. This points to a role for migratory birds in virus exchange between geographically distinct locations which has already been shown to be important for WNV and Usutu virus.

## Figures and Tables

**Figure 1 viruses-14-01476-f001:**
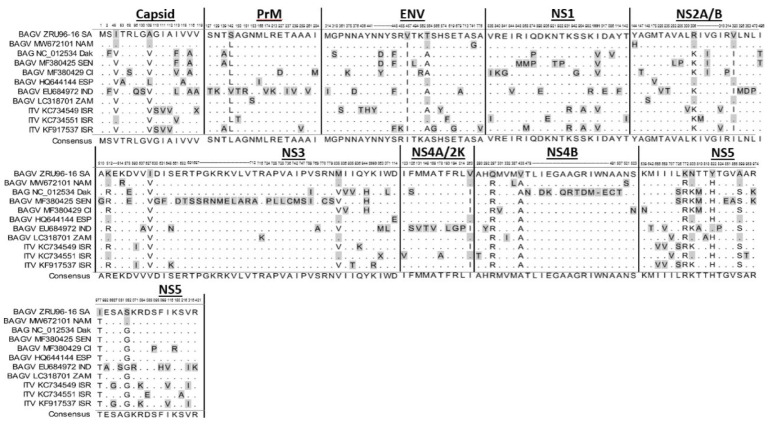
Multiple sequence alignment (MSA) of the polyprotein of selected BAGV and ITV isolates. Eight BAGV sequences, including ZRU96-16, and 3 ITV sequences were compared in an amino acid MSA across the length of the polyprotein. The accession number as well as the location of the isolate is depicted in the name of the sequence. Only amino acid sites in which variation was seen are depicted in the MSA. Amino acids highlighted in grey represent differences from the consensus (Genbank Accession no. MW463911).

**Figure 2 viruses-14-01476-f002:**
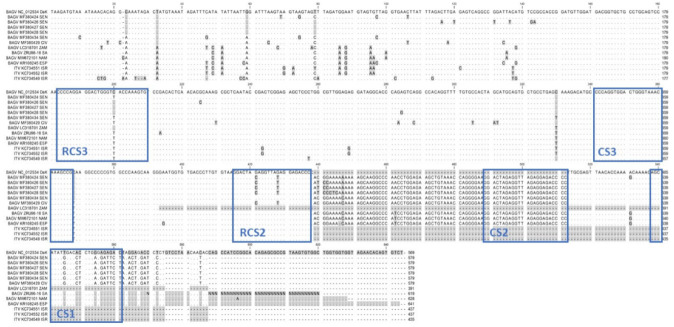
Multiple sequence alignment (MSA) of ITV and BAGV 3′UTR nucleotide sequences. The 3′UTR of ZRU96-16 was compared to a selection of BAGV and ITV 3′UTR segments. The boxes highlighted in blue represent conserved sequence motifs or repeat motifs (R) (CS1-3) that have been identified in other flavivirus UTRs. The region prior to the RCS3 and downstream the stop codon (TAA) is considered to be a variable region in most flaviviruses.

**Figure 3 viruses-14-01476-f003:**
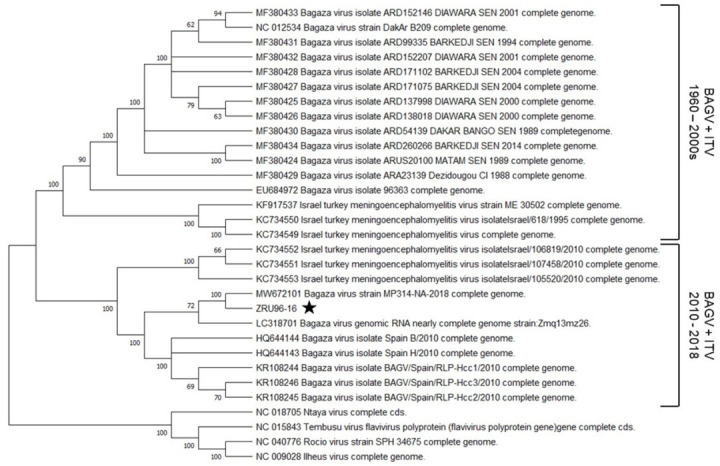
Maximum likelihood phylogenetic analysis of the Ntaya antigenic complex. The maximum likelihood method was used to infer evolutionary relatedness based on the general time reversible model (GTR) in Mega 7. A discrete Gamma distribution was used to model evolutionary rate with invariable rate variation at some sites (G+I). The tree with the highest log likelihood is shown. The tree was rooted on the Ntaya virus branch. Bootstrap support values are represented beside each node as a percentage of. The complete Genbank accession number is illustrated for each virus. The position of ZRU96-16 in the tree is highlighted by a black star.

## Data Availability

The full genome sequence reported in this study can be found on Genbank under the accession ID MW463911.

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
