# Peer review of "Phylogenetic Characterisation of the Full Genome of a Bagaza Virus Isolate from Bird Fatalities in South Africa"

_viruses, 2022, doi:10.3390/v14071476_

Round 1

Reviewer 1 Report

Mendes et al present a description of sequence data for a Bagaza virus isolated from the brain of a dead Himalayan monal pheasant. Obviously, the virus impacts bird life in outbreaks. Statistics of the numbers of birds affected in the different outbreaks should be presented in the Introduction. The sequence was determined using next generation sequencing to reveal a 132 nt truncation in the 3'-terminal end. Further, they presented phylogenetic analyses revealing that their isolate clustered with a recent clade of consisting of a Zambian strain as well as those from Spain and Turkey. Although the authors speculate on the 132 nt truncation, the discussion there is weak. Some RNA modeling could be done to compare the 3'terminal structures to other flaviviruses. How does the truncation contribute to species adaptation? Further, alignments presented indicate variability in NS5, an RdRP which is usually conserved. The authors should discuss this. Another aspect to be considered is the growth kinetics of the virus. The authors where able to isolate it from BHK-21 cells. How does it compare to other viruses?

Minor writing inconsistencies should be addressed in the text:

e.g. lines 86-88: please be consistent with manufacturer details and locations. In some cases this is omitted. Line 172: available in...

1uL should be 1 uL, or 30 min rather than 30mins. Please revise throughout the text as appropriate.

Author Response

Reply Reviewer 1

  • Statistics of the numbers of birds affected in the different outbreaks should be presented in the Introduction.

Line 46-47 added: “During this outbreak all 13 of the dead birds were found to be PCR positive for the virus”

Line 53-54 added: “ITV is severely neuropathogenic with mortality rates of between 15-30% and morbidity of approximately 80%“.

In line 61 added: “(a total of 8 PCR positive cases)…”

  • Some RNA modeling could be done to compare the 3'terminal structures to other flaviviruses.

Thank you for the suggestion.  We did initially try to do some basic mFold modelling of the UTRs, however we are not experienced in this are to interpret the changes to the structure.  As part of the revision for this manuscript, we carried out de novo assembly and found that the truncation of the sequence was resolved.  We have provided a new analysis of the 3’UTR sequences showing two clusters of the UTR according to where and when the virus was isolated. 

  • How does the truncation contribute to species adaptation?

As mentioned, the truncation was an artifact of the assembly method and thus is no longer relevant, however we have provided more detail on how UTRs contribute to replication.

Lines 275-278 added: “Specifically, mutations altering a small hairpin structure within the 3’UTR of DENV abrogated replication within insect cells with minimal effects in BHKs, illustrating the different requirements between hosts (26).” 

  • Further, alignments presented indicate variability in NS5, an RdRP which is usually conserved. The authors should discuss this

Thank you we have elaborated on the mutations found in NS5 and why we think that despite the usual sequence conservation, it is possible for some to happen.

Lines 206 – 208 added: “The amino acid mutations R2768K and T2973I within NS5 were unique to ZRU96-16, while the G3048S was only present on the South African and Namibian isolates.  This was of interest as NS5 is generally highly conserved”

Lines 334 to 340 added: “The amino acid variations within the NS5 protein were of interest, as generally this region is well conserved between Flaviviruses.  However, despite this we observed several locations in which variations were accommodated across the different BAGV and ITV sequences.  This may be explained by the amino acid forming part of a portion of the RNA-dependent-RNA polymerase which is not structurally constrained and/or the fact that there are still very few genomes sequences from these viruses and thus the full breadth of the diversity in this region has not been resolved.  This is exemplified by the fact that the G3048S variation would have been unique if not for the relatively new Namibian BAGV sequence”

  • Another aspect to be considered is the growth kinetics of the virus. The authors where able to isolate it from BHK-21 cells. How does it compare to other viruses?

We have not investigated the growth kinetics of BAGV.  A thorough investigation of the growth kinetics of this virus have been reported in Guggemos et al., 2021., in which the virus was grown in several cell lines to identify possible host susceptibility.  Since we do not have even half of the cell lines examined in this paper, we did not feel it would be relevant to add any growth kinetics data.  However, thank you for the suggestion as we do feel it will be interesting, should we succeed in culturing further isolates, from mosquitoes or dead animals, to compare their growth.

Minor writing inconsistencies should be addressed in the text:

e.g. lines 86-88: please be consistent with manufacturer details and locations. In some cases this is omitted.

Thank you this has been corrected

Line 172: available in...

This line has been deleted

1uL should be 1 uL, or 30 min rather than 30mins. Please revise throughout the text as appropriate.

Thank you.  The spaces have been added

Reviewer 2 Report

The manuscript of Mendes and co-authors describes the full genome sequencing and phylogenetic analysis of a Bagaza virus (BAGV) isolate obtained from a bird outbreak occurred in South Africa in 2016. This is an emerging vector-borne flavivirus which has increased its host and geographic range in the last twelve years, increasing at the same time the published scientific papers. Although it is a minor pathogen so far in terms of sanitary and economic consequences, surveillance and research should be reinforced to prevent and control further outbreaks or its potential zoonotic threat. Then, this work is of great interest for the animal health sector, and especially concerning the epidemiology of the disease and the evolution of the virus. The manuscript is in general well written and structured, with clear methodology and results sections, but with a short and limited discussion.

The present study is based on the previous one published in 2019 by the same authors (Steyn et al, Emerg Infect Dis. 2019, 25(12):2299-2302. doi: 10.3201/eid2512.190756), where they described the BAGV outbreaks occurred in South Africa in birds over 2016-2017. In this previous paper, the authors also performed a phylogenetic analysis of a partial region of the BAGV genome of several isolates obtained from these outbreaks. The present manuscript seems to be a continuation of this initial work, although the authors rarely name their previous publication only in the introduction section. On the other hand, the authors do not take into account in their study the recent paper published by Guggemos et al (Guggemos HD, Fendt M, Hieke C, Heyde V, Mfune JKE, Borgemeister C, Junglen S. Simultaneous circulation of two West Nile virus lineage 2 clades and Bagaza virus in the Zambezi region, Namibia. PLoS Negl Trop Dis. 2021 Apr 2;15(4):e0009311. doi: 10.1371/journal.pntd.0009311), which in my opinion could be very relevant for the results and conclusions of the submitted manuscript. In summary, although it is a very interesting study, I would recommend a major revision with the intention to improve the quality of the study and of the manuscript.

Specific comments:

-      - I would like to ask the authors to include their previous manuscript published in 2019 in the methodology and discussion sections, since this was the starting point for the present study. To what extent does the current study add value to the previous one? Was it not possible to fully sequence some more BAGV isolates from the 2016-2017 outbreaks?

-      - I strongly recommend to the authors to incorporate the BAGV genome sequence obtained from Namibia by Guggemos et al (see reference above) in their molecular analysis. In this mentioned publication, the authors report that Namibian BAGV strain is high closely related to the South African sequence (partial genome) obtained in 2016 by the authors of the present manuscript. I guess that the incorporation of the sequence from Namibia will provide a similar strong relationship that could refine the results and conclusions of the manuscript.

-       - I strongly suggest to the authors to perform a de novo assembly of the NGS reads added to mapping the reads to a BAGV reference sequence. If only the mapping strategy is used, the generated viral sequence is limited to the length of the reference sequence used and any potential extended length will be missed. Then, the genome size and the 3’ end truncation reported by the authors might not be accurate.  

-       - In the discussion section, I would like to suggest to incorporate some discussion regarding the potential role of the 3’ UTR in closest flaviviruses such as WNV. WNV is a member of the closely related Japanese encephalitis serocomplex of the flavivirus genus and, like in the case of BAGV, is transmitted by the same mosquito species and can produce a similar encephalitic disease in the same bird species.

Other minor comments:

-       - Please, be aware that Israel is not a European country but an Asian country, located in the Middle East region.

-       - Line 83: please, refer the publication of the pan-flavi PCR used in the study.

-       - Line 110: please, modify the final volume of the poly-U RNA reaction to 20 ml (7 ml of RNA+13 ml of reaction mix).

-       - Lines 177-178: Looking at the figure 1, NS5 region shows 3 unique sites, please correct or explain.

Author Response

Reply Reviewer 2

- I would like to ask the authors to include their previous manuscript published in 2019 in the methodology and discussion sections, since this was the starting point for the present study. To what extent does the current study add value to the previous one? Was it not possible to fully sequence some more BAGV isolates from the 2016-2017 outbreaks?

 Thank you for the suggestion.  Sections were added in the Materials and methods and the discussion in order to highlight the continuity of the work.

Lines 88 – 90 added: “The original identification of this specimen formed part of an initial study describing an outbreak of BAGV in pheasants on a private residence in Pretoria”

Lines 287-291 added: “Between 2016 – 2018, several Himalayan monal pheasants (Lophophoius impejanus) kept on private residences in Johannesburg succumbed to neurological infections from what was confirmed to be BAGV infections.  Phylogenetic analysis from partial NS5 gene regions showed that the outbreak was caused by a distinctive set of BAGV strains most closely related to Spanish BAGV”

We believe that a full genome sequence and the subsequent analysis of the sequence from an isolate derived from the outbreak (and the previous study) is a valuable addition to the literature as there are comparatively few genome sequences for BAGV and ITV.  What genome information there is available is also geographically constrained and thus the sequences from Namibia and South Africa, we believe are of value to analyse and report.  It took a significant amount of work to adapt one of these isolates to tissue culture and thus to have sufficient material to derive a full genome sequence.   Although, we did try, we were unable to adapt other isolates.  Previous attempts at deriving full genome sequences directly from the infected tissue were also unsuccessful.  Consequently, this was the only full sequence we could generate. 

-      - I strongly recommend to the authors to incorporate the BAGV genome sequence obtained from Namibia by Guggemos et al (see reference above) in their molecular analysis. In this mentioned publication, the authors report that Namibian BAGV strain is high closely related to the South African sequence (partial genome) obtained in 2016 by the authors of the present manuscript. I guess that the incorporation of the sequence from Namibia will provide a similar strong relationship that could refine the results and conclusions of the manuscript.

Thank you very much for this suggestion.  We were unaware of this publication and have incorporated the sequence into the analysis.  As predicted, the Namibian and South African sequences are very similar.  Furthermore, this sequence also clusters with more recent BAGV and ITV sequences, supporting the notion that there is continual exchange of these viruses between the Africa, the Middle East and Europe.

-       - I strongly suggest to the authors to perform a de novo assembly of the NGS reads added to mapping the reads to a BAGV reference sequence. If only the mapping strategy is used, the generated viral sequence is limited to the length of the reference sequence used and any potential extended length will be missed. Then, the genome size and the 3’ end truncation reported by the authors might not be accurate.

Thank you for this suggestion.  Again, you were proved correct in this prediction.  When we used a different de novo assembly tool (Genome Detective) we found that indeed we could resolve the 3’UTR.  We have subsequently changed the focus of the second section in the results (Lines: 231 – 286), in order to focus on sequence analysis of the UTR, which we still think is an interesting section of these genomes.  We found that the phylogenetic distinction between the sequences was maintained within the 3’UTRs in that West African sequences contained similar mutations within the variable region and southern African and Spanish sequences their own set of variations.  We also observed two different sets of CS1 sequence.   

-       - In the discussion section, I would like to suggest to incorporate some discussion regarding the potential role of the 3’ UTR in closest flaviviruses such as WNV. WNV is a member of the closely related Japanese encephalitis serocomplex of the flavivirus genus and, like in the case of BAGV, is transmitted by the same mosquito species and can produce a similar encephalitic disease in the same bird species.

Thank you, we have provided a general discussion on the role of the 3’UTR of the mosquito-borne flaviviruses, which to our knowledge all have similar structures and function similarly.  However, where possible we have also made reference to specific WNV studies and compared the BAGV UTR to those.  See lines 347-351 in the discussion.

Other minor comments:

-       - Please, be aware that Israel is not a European country but an Asian country, located in the Middle East region.

Thank you and noted.  We have changed sentences in which Israel is referenced as Europe.

-       - Line 83: please, refer the publication of the pan-flavi PCR used in the study.

References added.  Now line 88

-       - Line 110: please, modify the final volume of the poly-U RNA reaction to 20 ml (7 ml of RNA+13 ml of reaction mix).

Correct to 20 ul.  Now line 117.

-       - Lines 177-178: Looking at the figure 1, NS5 region shows 3 unique sites, please correct or explain.

There are 2 unique sites in NS5, since one of the sites previously noted is also present in the new Namibian sequence.  We have elaborated in the discussion section (Lines 334 – 341).  Essentially this could either be due to the fact that there does not have to be strict amino acid sequence conservation in NS5 and/or the fact that the full variation in NS5 amino acid sequences has not been explained due to so few sequences being available.

Round 2

Reviewer 1 Report

The authors' responses to this reviewer's concerns are noted. However, no attention was sufficiently paid to consistent writing in the methods section as highlighted before. The scientific abbreviation of minutes is "min" not "mins"

Line 364-5: no need to capitalize the common names of these birds.

Author Response

The authors' responses to this reviewer's concerns are noted. However, no attention was sufficiently paid to consistent writing in the methods section as highlighted before. The scientific abbreviation of minutes is "min" not "mins"

Thank you for pointing this out.  We have endeavored to go through our methods and correct all the “min” acronyms.  We have also added spaces between symbols where necessary.

Line 364-5: no need to capitalize the common names of these birds.

Thank you for the suggestion.  We have corrected the capitalization.

Reviewer 2 Report

I would like to thanks the authors for their extensive manuscript review which, in my opinion, has refined the results and conclusions of the study. Some modifications should be done before the acceptance of the manuscript:

1- Line 67: "To date 19 complete or near-complete BAGV genomes have been uploaded...". Please, update to 20 to include the Namibian sequence.

2- Figure 1 should be updated to incorporate the Namibia sequence to the alignment; if not, the information provided in the text does not match with the alignment shown in the figure.

3- Supplementary material should be modified. Table S1 should contain the newly genome annotation with the longer South Africa sequence; table S2 should include the data of the Namibian sequence.

Author Response

would like to thanks the authors for their extensive manuscript review which, in my opinion, has refined the results and conclusions of the study. Some modifications should be done before the acceptance of the manuscript:

Thank you for the helpful suggestions.

1- Line 67: "To date 19 complete or near-complete BAGV genomes have been uploaded...". Please, update to 20 to include the Namibian sequence.

Thank you.  Line updated.

2- Figure 1 should be updated to incorporate the Namibia sequence to the alignment; if not, the information provided in the text does not match with the alignment shown in the figure.

Thank you this was an oversight on our part.  We have updated Figure 1.

3- Supplementary material should be modified. Table S1 should contain the newly genome annotation with the longer South Africa sequence; table S2 should include the data of the Namibian sequence.

Thank you, the incorrect supplementary file was uploaded for the revision.  The correct file with this changes has been added